# Mental Health and Traumatization of Newly Arrived Asylum Seeker Adults in Finland: A Population-Based Study

**DOI:** 10.3390/ijerph18137160

**Published:** 2021-07-04

**Authors:** Ferdinand Garoff, Natalia Skogberg, Antti Klemettilä, Eero Lilja, Awa Ahmed Haji Omar, Olli Snellman, Anu E. Castaneda

**Affiliations:** 1Department of Public Health and Welfare, Finnish Institute for Health and Welfare, P.O. Box 30, 00271 Helsinki, Finland; natalia.skogberg@thl.fi (N.S.); antti.klemettila@thl.fi (A.K.); eero.lilja@thl.fi (E.L.); Awa.AhmedHajiOmar@thl.fi (A.A.H.O.); olli.snellman@migri.fi (O.S.); anu.castaneda@thl.fi (A.E.C.); 2Faculty of Medicine/Psychology, University of Helsinki, P.O. Box 21, 00014 Helsinki, Finland; 3Finnish Immigration Service, P.O. Box 10, 00086 Maahanmuuttovirasto, Finland

**Keywords:** asylum seekers, trauma, mental health

## Abstract

Asylum seekers frequently experience potentially traumatic events (PTEs), but the type and frequency vary depending on the country of origin. The cumulative effect of multiple PTEs and other stressors expose asylum seekers to a significant risk of mental ill health. The aim of the study was to examine the prevalence of PTEs, depression and anxiety symptoms, risk for psychological trauma, psychotropic medication use and previous mental health diagnoses among adult asylum seekers in the Asylum Seekers Health and Wellbeing (TERTTU) Survey (*n* = 784 respondents, participation rate 78.6%). A substantial majority (88.7%, 95% CI 86.9–90.3) of asylum seekers reported one or more PTEs before arriving to Finland. PTEs during the asylum-seeking journey were reported at 12.0% (95% CI 10.7–13.4), however, when examined by region of origin, the proportion was 34.5% (95% CI 29.5–39.8) for asylum seekers from Africa (excluding North Africa). Significant symptoms of depression were reported by 41.7% (95% CI 39.6–43.9) of asylum seekers and symptoms of anxiety by 34.2% (95% CI 32.1–36.2). Half of the asylum seekers were assessed as having at least a medium-risk for psychological trauma. Prevalence rates were higher among females and asylum seekers from Africa. This study highlights the importance of using screening tools to identify asylum seekers with severe mental health problems that may need referral to further assessment and treatment. Asylum seekers from Africa (excluding North Africa) should be given additional attention in initial health screenings and examinations.

## 1. Introduction

An asylum seeker is a person who has applied for international protection from another country and is waiting for a decision on their asylum application [1]. If granted asylum, the person will receive the legally recognized status of a refugee. There were 3.5 million asylum seekers worldwide at the end of 2018 [2]. Of these 664,405 were lodged in the 28 EU countries (64% males). The largest group of applicants were of Syrian nationality (84,235), followed by applicants of Afghan (47,755) and Iraqi (45,295) nationality [3]. The same year, 4548 asylum applications were submitted in Finland. Out of these, 2409 were first-time applications for international protection, whereas 2139 were applications for international protection by persons who had already received a final decision on a previous application [4].

Asylum seekers have frequently experienced potentially traumatic events (PTEs). PTE is an umbrella term that captures a host of adverse experiences, including violent experiences, but also other experiences such as separation from a family member or death of a family member due to an accident. Experiences of violence are common among forcibly displaced persons regardless of their country of origin. These experiences may include torture, witnessing death or injury, directly experiencing combat, and imprisonment or physical harm of loved ones [5]. Among asylum seekers these experiences may have occurred pre-migration, during migration or after resettlement [5,6,7]. The type and frequency of reported PTEs vary depending on the country of origin [8].

In addition, sex and age may contribute to the type and frequency of reported PTEs, with women more frequently reporting experiences of sexual violence [9], and older forcibly displaced persons reporting a greater accumulation of PTEs over time [10]. Multiple PTEs are often experienced in the same context or may be linked to each other, e.g., incarceration or kidnapping is often linked to violence and torture [11]. Both the number of PTEs experienced and length of exposure impact the mental health and well-being of forcibly displaced persons. The cumulative effect of multiple PTEs and other stressors expose forcibly displaced persons to significant risk of mental ill health [12,13].

In conflict and post-conflict settings, the impact of PTEs on mental health and well-being is often exacerbated by daily stressors [14] and ongoing stress [15]. Forcibly displaced persons that have experienced violence report increased mental health symptoms, including symptoms of post-traumatic stress disorder (PTSD), major depressive disorder (MDD) and anxiety. Somatic symptoms of distress are also frequently reported among forcibly displaced populations [16]. The impact of violence is persistent and affects the mental health of children, adolescents, and adults across the life span [5]. In addition to human suffering, this imposes a significant public health burden on countries receiving asylum seekers and refugees [17]. 

Refugees from conflict affected countries, e.g., Afghanistan, Syria, and Iraq, often report significant mental health symptoms when screened [8]. A meta-analysis encompassing studies published from 1990 to 2007 estimated that the prevalence rates among refugees were 44% for depression and 40% for anxiety disorders (including PTSD) [18]. In a more recent meta-analysis of mental illness in refugees and asylum seekers, estimates were slightly lower for depression at 32%, PTSD at 32% and anxiety disorders at 11% [19]. In a study from Sweden among refugees from Syria resettled in Sweden after 2011, 40% reported significant symptoms of depression, 32% anxiety disorder and 30% PTSD [20]. 

Population-based information about PTEs and mental health status of asylum seekers can help inform decision-making regarding allocation of resources for mental health interventions among asylum seekers from various countries of origin [8]. However, the generalizability of findings from previous studies on the health and well-being of asylum seekers have presented aggregate findings for all asylum seekers without distinguishing between different regions or countries of origin due to sample size restrictions and sampling methodology. Another common limitation in previous studies is that asylum seekers and other forced migrants are examined as one group. At the time of writing the authors have not been able to identify any previous studies among newly arrived asylum seekers that include a nationally representative population-based sample and with standardized objective health examination measures. The aim of this study is to examine PTEs and mental health of newly arrived asylum seekers in Finland by gender and region of origin. Mental health is examined through self-reported mental health symptoms, previous mental health diagnosis and the current regular use of psychotropic medications. 

## 2. Materials and Methods

### 2.1. The Study Population

This study is based on the Asylum Seekers Health and Wellbeing (TERTTU) Survey [21] conducted by the Finnish Institute for Health and Welfare in collaboration with the Finnish Immigration Service. The study population consisted of all newly arrived asylum seekers who applied for asylum in Finland for the first time (*n* = 2328) during the period of 19.2 to 30.11.2018. Asylum seekers were recruited into the study approximately 2 weeks after registration of their asylum application in Finland. Exclusion criteria were: (1) residence in a detention center; (2) previous application for asylum in another country and transfer to Finland based on international agreements (EU internal transfer); (3) having been returned to Finland according to the EU Dublin Regulation; (4) previous application for residence permit in Finland; (5) children born in Finland to asylum seekers; or (6) health reasons preventing participation in the study reported by reception center staff. After exclusion criteria were applied, *n* = 674 were excluded. 

The population-based sample was drawn on a weekly basis from an electronic database maintained by the Finnish Immigration Services containing data on all asylum seekers in Finland. There were two breaks in the data collection (30.4–13.5.2018 and 27.8–9.9.2018), the first due to issues with the Finnish Immigration Service providing the necessary data and the second due to sick leaves and absences of field work personnel (*n* = 158 excluded). In addition, there was an IT error during the summer that lead to a sampling error (*n* = 63 excluded). The final sample of the TERTTU Survey consisted of *n* = 1433 first time asylum seekers. The current study is based on the adult sub-sample of persons aged 18 years and older (*n* = 998) with a participation rate of 78.6% (*n* = 784).

### 2.2. Data Collection

The sampling and data collection processes are summarized in the Appendix A. A more detailed description of the survey methods can be found in the study protocol [21]. Data collection of the TERTTU Survey was implemented by eight trained multilingual research nurses. If the mother tongue of the study participant was another language than those spoken by the research nurses, a professional interpreter from an accredited company was used. The professional interpreters were briefed on the importance of following the standardized study protocol. The initial contact between the research nurse and persons belonging to the study sample were facilitated by the reception center personnel because asylum seekers could not be directly contacted by the nurse. Participants were informed about the voluntary nature of the study and participation was based on written informed consent. The study consisted of a health examination and a face-to-face interview, altogether these lasted 1.5–2 h.

The demographic data on the participants in this study are presented in Table 1. The asylum seekers in the study were classified into four geographical regions of origin according to their citizenship: (1) Russia and the former Soviet Union (Russia and FSU), which included mainly citizens of the Russian Federation (78%); (2) Middle East and North Africa (MENA), which included mainly of citizens of Turkey (29%), Iran (24%) and Iraq (18%); (3) Africa excluding North Africa (Africa), included mainly of citizens of Somalia (20%), Nigeria (17%), Angola (13%) and Cameroon (12%); and (4) Other regions, which included citizens of Nicaragua (14%), Albania (12%), Bangladesh (7%), India (5%), Cuba (5%), Kosovo (5%) and Sri Lanka (5%). The largest group of asylum seekers were those from the MENA region, followed by Russia and FSU. Asylum seekers from Africa were younger than those from other geographical regions. 

### 2.3. Measures

Data reported in this paper include sociodemographic characteristics (gender, age group and geographical region of origin), self-reported psychotropic medication, self-reported previous mental health diagnosis, PTEs, and mental health symptoms based on two mental health-screening tools. Data were collected during face-to-face interviews conducted by research nurses or using self-administered questionnaires. 

PTEs were investigated using 10 items with an additional clarification if these PTEs happened before or during the asylum-seeking journey. The items were adapted from the Harvard Trauma Questionnaire [22]. The three response options were: (a) no, (b) yes, before the journey and (c) yes, during the journey. Both yes options could be selected. The questionnaire included the following PTEs: being in combat situations, natural disaster, seeing a violent injury or death, being the target of physical harm (prevented from moving, pushing or shoving), being the target of severe physical violence (hitting, kicking, strangling, use of a weapon), being imprisoned or kidnapped, being tortured (with a definition based on the UNCAT [23]), being the target of sexual violence, being forced or cheated into doing something that you did not want to do. 

The Hopkins Symptom Checklist-25 (HSCL-25) [24] was used to detect symptoms of anxiety and depression. The scale includes a 10-item subscale for anxiety and a 15-item subscale for depression, with each item scored in terms of frequency from 1 (not at all) to 4 (often). HSCL mean scores (ranging from 1.0–4.0) were calculated for each case. The recommended cut-off of 1.75 was used for significant depression and/or anxiety symptoms [25,26]. Additionally, the prevalence rates of suicidality as assessed by the HSCL item “thoughts of ending your life” were examined, as this is an indication for mental health referral. 

The PROTECT Questionnaire (PQ) [27] was used as a screening tool for identification of asylum seekers that may have experienced psychological trauma. The PQ consists of items concerning the frequency of symptoms of PTSD (4 items: nightmares, anger, thinking about painful past events, feeling scared or frightened); symptoms of both MDD and PTSD (4 items: problems falling asleep, forgetting things, losing interest in things, trouble concentrating); and pain symptoms (2 items: headaches, other physical pains). The answer options were yes/no. The number of items for which “yes” were reported were added together to indicate the risk for post-traumatic mental health problems (low-risk 0–3 yes answers; medium-risk 4–7 yes answers; high-risk 8–10 yes answers). Referral to a mental health professional is recommended for those in the medium- and high-risk categories. 

The interview also included a question about previous depression or other mental health diagnosis given by a medical doctor (yes/no), as well as a question about current mental health medication (yes/no), with the possibility to specify the type of medication (yes/no; tranquilizer, medication for depression, sleep medication).

### 2.4. Statistical Methods

The results were examined as proportions and their 95% confidence intervals (CI). The CIs were calculated using logit transformation and finite population correction [28] was applied. The random forest method [29] was used to estimate the participation probability out of those who were reached in the data collection (*n* = 1322). In the calculation, we used register information on sex, age, country of origin and place of residence. The inverse of these probabilities were then used as sample weights in all analyses in the article [29]. Additionally, all regional results were adjusted for age and sex with logistic regression using predictive margins [30]. The sample weights were calculated using the randomForest package in R. All other analyses were performed with the SAS-callable SUDAAN 11.0.3 software.

## 3. Results

### 3.1. Experiences of PTEs 

Table 2 summarizes PTEs by region of origin. The majority of respondents reported that they had experienced at least some PTEs. Most experiences of violence were significantly more frequently reported by males, but sexual violence was more frequently reported by females. The PTEs reported also varied by region of origin. Almost all asylum seekers from Africa reported PTEs. The rates of reported sexual violence were more than three times those of other regions with more than half of females reporting sexual violence. 

See Appendix A for summaries of PTEs reported by asylum seekers before and during the asylum-seeking journey. Reported PTEs during the asylum-seeking journey (12.0%, *n* = 92, 95% CI 10.7–13.4) were significantly less common than those prior to the asylum-seeking journey. PTEs during the asylum-seeking journey were significantly more common among asylum seekers from Africa, as reported by a third (34.5%, *n* = 46, 95% CI 29.5–39.8) and less common among asylum seekers from other regions (Russia and FSU 2.2%, *n* = 5, 95% CI 1.3–3.7; MENA 11.6%, *n* = 37, 95% CI 9.6–13.8; and Other 4.4%, *n* = 4).

### 3.2. Mental Health Status

Table 3 summarizes reported mental health symptoms, medication use and previous diagnoses by gender and region of origin. One third of asylum seekers reported significant anxiety symptoms, whereas depression symptoms were slightly more common. One third were assessed as being at medium-risk and one sixth at high-risk for psychological trauma. Females reported higher levels of depression and anxiety symptoms and were more often classified to be at a high-risk for psychological trauma than males. Asylum seekers from Africa reported more depression and anxiety symptoms than asylum seekers from other regions. Current regular use of psychotropic medication and prevalence of previous mental health diagnosis by a medical doctor were low. 

## 4. Discussion

Most adult asylum seekers reported one or more PTEs before arriving in Finland (88.7%), which is consistent with findings from previous studies [5,6,7]. The most commonly reported PTEs were physical harm and violence. Torture was also reported frequently, particularly among males. This is similar to findings from a previous study conducted in Finland [31]. Generally, males reported more violence-related PTEs than females. Experiences of combat situations were as common among males and females, whereas sexual violence was more frequently reported by females. The prevalence rates for sexual violence among female asylum seekers was close to that reported in previous studies [9]. Our findings support the previous evidence that asylum seekers are exposed to a significant burden of PTEs and that all asylum seekers should be considered at increased risk to develop mental health disorders and other adverse conditions [10]. However, as most at-risk individuals will not develop mental health problems, it is important to identify high-risk sub-populations in order to target interventions efficiently [32], particularly as positive psychological changes are possible even while struggling with mental health symptoms [33].

Regional differences in both the frequency and type of reported PTEs were significant. Asylum seekers from Africa reported most types of PTEs more frequently than asylum seekers from other regions, except for natural disasters. Sexual violence was reported by more than half of female asylum seekers from Africa and more than a quarter reported experiencing sexual violence during their asylum-seeking journey. Slightly more than a tenth of asylum seekers reported PTEs during their asylum-seeking journey. This figure was lowest among asylum seekers from Russia or FSO, this may be due to the fact that their asylum-seeking journey to Finland is short. PTEs both before and during the asylum-seeking journey were more frequently reported among asylum seekers from Africa. Females from Africa reported sexual violence and being forced or cheated into doing something that they didn’t want to do during the asylum-seeking journey more frequently than males. This may indicate a link to human trafficking, which is frequent among asylum seekers from Africa [34]. In Western Europe most victims of trafficking are migrants. Therefore, improving measures for identification and protection of victims of trafficking among asylum seekers, as well as improving granularity and coverage of data collection are recommended [34].

The estimates of significant depression and anxiety symptoms among asylum seekers in this study were similar to those reported in previous studies among refugees [18]. However, it is worth noting mental health is likely to deteriorate during extended periods of visa insecurity [35], therefore mental health symptoms are likely to be more prevalent among those in prolonged application processes or reapplying for asylum compared to the newly arrived asylum seekers in this study. Females reported significantly more symptoms of depression and anxiety than males, this is consistent with previous studies [8,20]. In the current study, asylum seekers from Africa reported these symptoms significantly more frequently than those from other regions, including suicidal thoughts. Half of the asylum seekers were assessed as being at medium- or high-risk for having experienced psychological trauma. Females were more likely to belong to the high-risk group than males. Based on the findings in this study and following the recommendations set out in the PQ screening tool, half of asylum seekers should be referred to mental health professionals for a more in-depth assessment of their symptoms [27]. However, asylum seekers in this study reported much fewer symptoms on the PQ than in a previous validation study with asylum seekers in Germany, where two thirds of participants were classified as high-risk on the PQ [36]. One explanation for this difference may be that mental health symptoms are under-reported in interview situations, particularly if an interpreter is needed [37]. On the other hand, this study is a population-based study whereas the previous study may have had a greater degree of selection bias as participants were approached in a less systematic fashion. Effective screening and identification of persons in need of services may improve the immediate and long-term health of asylum seekers [38,39]. Referral should be made as soon as possible as delayed treatment may increase the risk of developing mental health disorders [40]. In addition, it is important to address any barriers to accessing necessary services [41].

Despite the observed high prevalence of mental health symptoms, few of the asylum seekers in this study reported current, ongoing use of mental health medication or previous mental health diagnoses. This may be a consequence of poor availability of mental health services in many countries from where asylum seekers originate [42,43].

In the light of the findings of this study additional studies are warranted to identify the overlap between symptom profiles [44] and regions of origin among asylum seekers. In addition, further validation studies of the PQ measure are necessary. 

### Strengths and Limitations

A significant strength of this study is the representative nature of the data, using large-scale population-based survey data encompassing the majority of newly arrived asylum seekers to Finland in 2018. A further strength is the comprehensive data collection method, which allows for identifying groups particularly at risk by examining differences by gender, and region of origin. However, the population differed from that of other countries in Europe, the largest number of asylum seekers in this study were from the Russian Federation (28%) and Turkey (15%) while asylum seekers from Syria and Afghanistan were the most common countries of origin EU countries in 2018 [3,45]. The cultural diversity of the asylum seekers in this study may also contribute to experienced PTEs, reported mental health symptoms and access to treatment. This limits the generalizability of findings. The gender distribution was similar to that in the other EU countries. 

The data for this study were collected in 2018. Since then the COVID-19 pandemic has changed the situation for those seeking international protection. The number of applications for international protection in the EU had stabilized to about two thirds of pre-COVID-19 levels by March 2021 [46]. The decline in applications was ascribed to closed borders and reduced mobility, but the number of asylum seekers may increase if restrictions on mobility are relaxed.

A strength of the study was that standardized and validated measures for mental health symptoms were used, which allows for comparison with previous studies [25,26,36]. However, the assessment of PTEs in this study is limited to reporting the prevalence of individual types of PTEs. Further studies should explore the accumulation of traumatic experiences, as well as the frequency, severity and duration of PTEs experienced. 

Mental health symptoms are often under-reported in surveys as severe mental health symptoms and disorders are often considered stigmatizing for the individual and disclosure may have spillover effects on friends or family [47,48]. On the other hand, asylum seekers have an incentive to report their symptoms as evidence of persecution. Sexual violence is also often under-reported [9,49], particularly in face-to-face interviews [50]. To address this in this study an option was available to answer sensitive questions, including the HSCL and PQ, using self-report questionnaires. The majority (66%) of asylum seekers used this option. These questions were asked in the interview if the participant was illiterate or if an interpreter was used. The use of professional interpreters in this study may have improved detection of mental health symptoms and PTEs through language concordance and improved communication quality [51]. While conducting the study in the mother tongue of the participants is a significant strength of this study, as discussed above, it is also possible that some persons may have been more reluctant to discuss openly all of their previous experiences and current mental health symptoms especially if these have a degree of stigmatisation in their country of origin or culture. To address this potential issue, research nurses were trained in how to build a trusting and safe environment for the participants. Participants were also assured about the confidentiality of all of the information they provided.

Another limitation of the study is that self-reported previous mental health diagnosis may not accurately reflect previous mental health diagnoses given by a doctor as self-reported diagnoses are prone to biases [52]. As with mental health symptoms, biases in reporting mental health diagnoses in this study may include under-reporting due to mental health related stigma or over-reporting in order to exaggerate vulnerability in the asylum-seeking process. However, in some populations self-reported mental health diagnoses have adequate validity [53], but this is likely to be quite context sensitive. This limitation does not however significantly undermine the general finding of this study that the rates of self-reported previous mental health diagnosis by a doctor among asylum seekers are low compared to lifetime prevalence rates of mental health disorders, particularly for conflict affected populations [54].

## 5. Conclusions

Most of the asylum seekers reported one or more PTEs before arriving in Finland. PTEs were significantly more often reported by males than females, with the exception of sexual violence. Prevalence of depression and anxiety symptoms were high and half of the respondents were assessed to have a medium- to high-risk for trauma-related mental health symptoms. Mental health symptoms were more common among females than males. Asylum seekers from Africa (excluding North Africa) reported higher rates of PTEs both in the country of origin and during the asylum-seeking journey than asylum seekers from other regions. They also had higher prevalence rates of depression and anxiety symptoms and were at a higher risk of trauma-related mental health symptoms. This study highlights the importance of using screening tools to identify asylum seekers with severe mental health problems that may need referral to further assessment and treatment. Asylum seekers from Africa (excluding North Africa) should be given additional attention in initial health screenings and examinations.

## Figures and Tables

**Table 1 ijerph-18-07160-t001:** Demographic data.

Participants	Gender		Age Group			
	Male	Female	18–29	30–39	Over 39	Total
Region of Origin	N (%)	N (%)	N (%)	N (%)	N (%)	N (%)
Russia and FSU	135 (59.0)	94 (41.0)	71 (31.0)	73 (31.9)	85 (37.1)	229 (29.2)
MENA	208 (60.8)	134 (39.2)	123 (36.0)	140 (41.0)	79 (23.1)	342 (43.6)
Africa	75 (58.1)	54 (41.9)	63 (48.8)	51 (39.5)	15 (11.6)	129 (16.5)
Other	55 (65.5)	29 (34.5)	36 (42.9)	27 (32.1)	21 (25.0)	84 (10.7)
Total	473 (60.3)	311 (39.7)	293 (37.4)	291 (37.1)	200 (25.5)	784 (100)

**Table 2 ijerph-18-07160-t002:** Age-adjusted prevalence of potentially traumatic events (PTEs) by region of origin.

Reported PTEs	Region of Origin				Total
	Russia/FSU	MENA	Africa (excl. NA)	Other	
	Total *n* = 229	Total *n* = 332	Total *n* = 125	Total *n* = 83	Total *n* = 769
	% (CI)	% (CI)	% (CI)	% (CI)	% (CI)
Combat situations					
Male	20.3 (16.7–24.5)	31.8 (28.1–35.8)	48.0 (41.2–54.8)	35.1 (27.7–43.3)	31.4 (29.0–34.0)
Female	13.7 (10.1–18.3)	28.1 (23.5–33.1)	47.8 (39.6–56.1)	42.7 (32.0–54.2)	28.9 (25.9–32.1)
Total	17.6 (14.9–20.7)	30.3 (27.4–33.4)	48.0 (42.8–53.4)	37.9 (31.6–44.5)	30.4 (28.5–32.4)
Natural disaster					
Male	14.9 (11.6–18.8)	26.9 (23.5–30.7)	12.5 (8.7–17.7)	23.7 (17.4–31.4)	20.7 (18.6–23.0)
Female	11.4 (8.2–15.8)	18.7 (15.2–22.9)	19.9 (14.0–27.6)	32.6 (23.0–43.9)	18.5 (16.0–21.2)
Total	13.4 (11.0–16.2)	23.6 (21.0–26.4)	15.9 (12.3–20.3)	27.0 (21.5–33.3)	19.8 (18.2–21.6)
Seeing violent injury or death					
Male	52.5 (47.5–57.5)	62.0 (57.9–65.9)	84.3 (78.8–88.6)	66.7 (58.7–73.8)	63.5 (60.8–66.0)
Female	33.9 (28.5–39.7)	35.7 (30.8–40.9)	72.9 (64.9–79.6)	57.4 (45.9–68.1)	43.8 (40.5–47.2)
Total	45.0 (41.2–48.8)	51.3 (48.2–54.5)	79.7 (75.2–83.6)	62.6 (56.0–68.7)	55.5 (53.4–57.6)
Physical harm					
Male	78.0 (73.6–81.9)	50.2 (46.1–54.3)	83.3 (77.1–88.0)	64.6 (56.6–71.8)	65.0 (62.4–67.5)
Female	52.5 (46.6–58.4)	31.1 (26.4–36.1)	71.3 (62.5–78.7)	42.8 (31.9–54.5)	45.3 (41.9–48.7)
Total	67.6 (64.0–71.0)	42.5 (39.3–45.6)	78.5 (73.5–82.8)	55.7 (49.2–62.0)	57.0 (54.9–59.1)
Physical violence					
Male	69.3 (64.5–73.7)	47.3 (43.2–51.4)	82.9 (77.2–87.4)	59.4 (51.3–67.0)	60.4 (57.8–63.0)
Female	31.8 (26.5–37.5)	25.8 (21.5–30.5)	79.9 (72.2–85.9)	36.6 (26.3–48.2)	38.1 (34.8–41.5)
Total	53.8 (50.2–57.5)	38.5 (35.5–41.7)	82.0 (77.6–85.7)	50.0 (43.6–56.4)	51.3 (49.2–53.4)
Imprisoned or kidnapped					
Male	42.3 (37.5–47.3)	32.2 (28.4–36.2)	61.4 (54.6–67.8)	34.8 (27.6–42.6)	40.0 (37.4–42.7)
Female	11.6 (8.3–15.9)	20.6 (16.6–25.2)	45.9 (37.8–54.1)	29.1 (19.1–41.7)	23.1 (20.2–26.2)
Total	29.7 (26.5–33.2)	27.5 (24.7–30.4)	55.3 (50.1–60.4)	32.1 (26.1–38.8)	33.1 (31.1–35.2)
Torture					
Male	45.9 (40.9–50.9)	44.1 (40.0–48.2)	67.3 (60.7–73.3)	52.8 (44.9–60.5)	49.2 (46.5–52.0)
Female	7.3 (4.7–11.0)	33.1 (28.4–38.1)	55.5 (47.4–63.4)	37.6 (27.0–49.6)	29.9 (26.8–33.2)
Total	29.8 (26.6–33.3)	39.5 (36.4–42.7)	62.7 (57.6–67.6)	46.4 (40.0–52.9)	41.3 (39.2–43.4)
Sexual violence					
Male	5.8 (3.9–8.7)	6.8 (5.0–9.3)	18.2 (13.5–24.1)	3.5 ^1^	7.8 (6.5–9.4)
Female	14.8 (11.1–19.4)	16.9 (13.5–20.9)	56.4 (47.9–64.5)	24.0 (15.5–35.1)	24.0 (21.2–27.0)
Total	9.5 (7.5–11.9)	11.0 (9.1–13.1)	34.2 (29.6–39.1)	11.8 (8.1–16.9)	14.4 (13.0–16.0)
Forced or cheated					
Male	37.2 (32.5–42.1)	33.4 (29.6–37.4)	50.5 (43.7–57.3)	40.6 (33.0–48.6)	37.6 (35.0–40.2)
Female	25.9 (21.1–31.4)	29.8 (25.3–34.8)	69.0 (60.9–76.1)	23.5 (15.2–34.6)	35.2 (32.0–38.6)
Total	32.5 (29.0–36.1)	32.0 (29.0–35.1)	58.7 (53.3–63.8)	33.9 (28.0–40.3)	36.6 (34.6–38.7)
Any PTE					
Male	86.4 (82.6–89.5)	87.9 (84.9–90.3)	97.3 ^1^	85.5 (79.1–90.2)	88.7 (86.9–90.3)
Female	72.7 (67.1–77.6)	67.6 (62.5–72.2)	87.0 (79.6–91.9)	82.7 (73.8–89.1)	73.7 (70.6–76.6)
Total	80.8 (77.6–83.6)	79.5 (76.8–82.0)	93.0 (89.5–95.4)	84.1 (78.9–88.2)	82.6 (81.0–84.2)

FSU, former Soviet Union; MENA, Middle-East and North Africa; Africa (excl. NA), Africa excluding North Africa; CI, confidence interval. **^1^** Confidence interval cannot be calculated due to low or high frequency.

**Table 3 ijerph-18-07160-t003:** Age-adjusted prevalence of mental health symptoms, medication use and previous mental health diagnoses by region of origin.

Mental Health Measures	Region of Origin				
	Russia/FSU	MENA	Africa (excl. NA)	Other	Total
	Total *n* = 220	Total *n* = 318	Total *n* = 116	Total *n* = 82	Total *n* = 736
	% (CI)	% (CI)	% (CI)	% (CI)	% (CI)
Depression symptoms					
Male	25.7 (21.6–30.3)	36.5 (32.6–40.7)	61.3 (54.3–67.9)	37.4 (30.0–45.6)	37.7 (35.0–40.4)
Female	39.3 (33.6–45.3)	47.3 (42.0–52.5)	69.1 (60.5–76.5)	43.6 (32.8–55.0)	47.5 (44.1–51.0)
Total	31.2 (27.8–34.9)	40.9 (37.7–44.2)	64.4 (59.0–69.5)	40.1 (33.8–46.7)	41.7 (39.6–43.9)
Anxiety symptoms					
Male	23.7 (19.7–28.2)	26.8 (23.2–30.7)	41.0 (34.3–48.1)	31.0 (24.0–39.0)	28.9 (26.4–31.5)
Female	41.6 (35.8–47.5)	39.0 (34.0–44.2)	51.1 (42.2–60.0)	43.3 (32.5–54.8)	41.8 (38.4–45.2)
Total	31.0 (27.6–34.7)	31.7 (28.7–34.9)	45.0 (39.5–50.7)	36.1 (30.0–42.7)	34.2 (32.1–36.2)
Suicidal thoughts					
Male	3.5 (2.1–5.8)	4.7 (3.2–6.9)	19.5 (14.0–26.4)	10.2 (6.1–16.6)	7.2 (5.8–8.8)
Female	2.2 ^1^	8.5 (6.0–12.1)	20.8 (14.5–28.9)	2.9 ^1^	7.8 (6.1–9.9)
Total	2.9 (1.9–4.5)	6.2 (4.8–8.1)	20.0 (15.6–25.2)	7.6 (4.7–12.0)	7.4 (6.3–8.7)
HSCL Sum score					
Male	26.1 (21.9–30.7)	31.7 (28.0–35.8)	59.0 (51.9–65.8)	31.9 (24.9–39.9)	34.7 (32.1–37.4)
Female	40.7 (35.1–46.6)	45.5 (40.3–50.8)	60.4 (51.3–68.9)	43.7 (33.0–55.1)	45.7 (42.2–49.1)
Total	32.1 (28.6–35.7)	37.3 (34.2–40.5)	59.3 (53.6–64.8)	36.7 (30.6–43.2)	39.2 (37.1–41.3)
	Total *n* = 228	Total *n* = 332	Total *n* = 124	Total *n* = 83	Total *n* = 767
Risk for traumatization	% (CI)	% (CI)	% (CI)	% (CI)	% (CI)
Low					
Male	66.5 (61.6–71.1)	51.8 (47.7–55.9)	39.7 (33.4–46.5)	58.0 (49.9–65.6)	54.5 (51.8–57.2)
Female	54.8 (48.9–60.6)	34.9 (30.2–39.9)	34.5 (27.0–42.9)	43.9 (33.5–54.9)	42.0 (38.8–45.4)
Total	61.8 (58.0–65.4)	44.9 (41.8–48.1)	37.7 (32.8–43.0)	52.3 (45.8–58.6)	49.4 (47.3–51.5)
Medium					
Male	27.1 (22.9–31.8)	31.9 (28.1–35.8)	45.9 (39.2–52.7)	30.2 (23.3–38.2)	32.8 (30.3–35.4)
Female	32.5 (27.2–38.3)	40.1 (35.2–45.3)	35.7 (28.2–43.9)	38.2 (27.8–49.9)	36.7 (33.5–40.1)
Total	29.3 (26.0–33.0)	35.2 (32.2–38.4)	41.4 (36.3–46.7)	33.4 (27.5–40.0)	34.4 (32.4–36.5)
High					
Male	6.4 (4.4–9.1)	16.3 (13.4–19.7)	14.4 (9.9–20.4)	11.8 (7.4–18.3)	12.7 (10.9–14.7)
Female	12.7 (9.2–17.2)	25.0 (20.7–29.8)	29.8 (22.6–38.2)	17.9 (11.2–27.2)	21.2 (18.5–24.2)
Total	8.9 (7.0–11.3)	19.8 (17.3–22.6)	20.8 (16.6–25.7)	14.3 (10.3–19.6)	16.2 (14.6–17.8)
	Total *n* = 218	Total *n* = 323	Total *n* = 121	Total *n* = 81	Total *n* = 743
Psychotropic medication	% (CI)	% (CI)	% (CI)	% (CI)	% (CI)
Male	5.7 (3.8–8.5)	7.1 (5.3–9.5)	6.8 ^1^	5.6 ^1^	6.6 (5.4–8.1)
Female	7.9 (5.1–11.9)	12.1 (8.9–16.4)	10.0 ^1^	6.7^1^	9.7 (7.7–12.2)
Total	6.6 (4.9–8.8)	9.1 (7.3–11.2)	8.4 (5.5–12.7)	6.0^1^	7.9 (6.7–9.2)
	Total *n* = 229	Total *n* = 333	Total *n* = 125	Total *n* = 83	Total *n* = 770
Previous mental health diagnosis	% (CI)	% (CI)	% (CI)	% (CI)	% (CI)
Male	9.6 (7.2–12.8)	10.5 (8.3–13.2)	7.5 (4.5–12.2)	14.2 (9.2–21.2)	10.2 (8.7–12.0)
Female	15.5 (11.7–20.2)	22.6 (18.4–27.3)	9.4 ^1^	18.3 (10.7–29.7)	17.8 (15.3–20.7)
Total	12.0 (9.8–14.7)	15.4 (13.2–17.9)	8.3 (5.6–12.1)	16.0 (11.5–21.8)	13.3 (11.9–14.9)

FSU, former Soviet Union; MENA, Middle-East and North Africa; Africa (excl. NA), Africa excluding North Africa; CI, confidence interval. **^1^** Confidence interval cannot be calculated due to low frequency.

## Data Availability

Data available on request upon an accepted research proposal.

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
