# Peer review of "Mental Health and Traumatization of Newly Arrived Asylum Seeker Adults in Finland: A Population-Based Study"

_ijerph, 2021, doi:10.3390/ijerph18137160_

Round 1

Reviewer 1 Report

The manuscript entitled “Mental health and traumatization of newly arrived asylum seeker adults in Finland: A population-based study” presents interesting issue, but some areas should be corrected.

Major:

Authors present the results which were gathered in 2018. Unfortunately such results, especially associated with international transport and mental heath, are in 2021 out of date, as both issues are changed because of COVID-19. Authors should be aware, that the COVID-19 issue is a novel mental health problem, and that in many countries the international transport is seriously limited, so asylum seekers experience a novel problems. When we are facing COVID-19 issue, the description of the general situation observed in 2018 is currently of a low scientific value, as we have now novel situation and novel problems. Taking this into account, Authors should present a proper perspective for their results. The results may be valuable, but should be presented as those gathered before COVID-19 pandemic. At the same time, Authors should introduce potential influence of COVID-9 pandemic on the situation observed based on their results.

Abstract:

Instead of indicating only what was done (“data […] was used to examine…”), Authors should clearly specify the aim of their study (e.g. “The aim of the study was…”).

Introduction:

Authors should prepare this section not only to be interesting for Finnish readers, but to be interesting for international readers. If Authors prepare their manuscript only for their national readers, they should publish it in some national journal. So, Authors should present here international data from various countries, not only the Finnish/ Swedish ones.

Authors should present in this section major countries of origin for current asylum seekers (not only in the case of Finland, but also other countries).

Materials and Methods:

Authors introduce their data as based on a representative sample, but the representativeness of the studied sample was not verified. Authors should verify it based on the country of origin, gender and age and indicate if they are representative for the general population of asylum seekers for Finland/ Europe/ global population (not only for 2018 year, but also in general) in order to indicate if this population is in fact representative or not. If it is not representative, Authors should indicate it as a limitation of the study.

Results:

Results which are presented in tables should not be extensively described in the text

Discussion:

This section should be deepened, to present not only the current situation, but its possible implications.

Authors should broadly present limitations of the study (see above).

Conclusions:
this section should be brief, without references and should present major observations from the presented study (2-3 simple sentences based directly on the presented study to present major results and their implications are enough).

Author Response

To the editors and reviewers,

Thank you for taking the time to review our paper. We appreciate the valuable comments and have done our best to incorporate these into the revised manuscript. We have addressed each of the points in the revised manuscript and provide a summary in the attached document.

The authors

Reviewer 2 Report

This is a generally useful report on the plight of newly arrived adult asylum-seekers in Finland.

Specific comments:

  1. Please change "newly arrived asylum seeker adults" to "newly arrived adult refugees and asylum seekers".
  2. Did the study also include refugees? It seems so to me as no distinction was made between whether forcibly displaced or not. In fact, the definition cited by the authors seems incomplete. An asylum seeker is someone who is seeking international protection but whose claim for refugee status has not yet been determined. In contrast, a refugee is someone who has been recognized under the 1951 Convention relating to the status of refugees to be a refugee.
  3. On average, how long did an interview take?
  4. When presenting results, please provide the exact n and accompanying percentage, e.g. "Reported PTEs during the asylum-seeking journey (12.0%, n=?, 95% CI 10.7–13.4)".
  5. "Suicidality was reported by seven percent of the respondents.Suicidality was significantly" - please note the lack of spacing.
  6. The authors use the word “suicidality”, which has long fallen out of favour with psychiatrists and psychologists due to its imprecision. Are you referring to suicidal thoughts or suicide attempts? Please be specific.
  7. In the discussion section or under areas for future work, authors can also briefly discuss posttraumatic growth. Beyond anxiety, grief and depression, self-reported positive psychological changes, or posttraumatic growth is possible, albeit people simultaneously struggled with mental illness, as seen in previous studies of Hurricane Katrina survivors (citation: pubmed.ncbi.nlm.nih.gov/32943541). 
  8. "Data available on request upon an accepted research proposal" - what does this mean? The underlying data should be made publicly available. If this was not possible, please provide a reason why.

Author Response

(The authors gave the same response as above.)

Reviewer 3 Report

This is population-based study on the prevalence of PTEs among the asylum seekers from different countries of origin.  The study identifies important profile characteristics of persons with PTEs.  Limitations of the study were briefly noted.  The study results have confirmed the fact that stressful events or life-change events are likely to predispose the subjects to mental health problems.

Minor amendments for improving the clarity of the report are suggested as follows:

  1. The Severity of PTEs Experienced:  Is there a way to determine the severity of the event (life threatening, stress producing, etc.) and the duration (the frequency or intensity of the event reported)?
  2. Potenial Mediators:  The absence of social support or family support is a mediator factor between stressful event and PTEs or mental health).  The literature has clearly documented that the important factors such as the level of social support (instrumental and emotional support) available may directly or indirectly affect the mental wellbeing.
  3. Sample Clarification:  Because of a large number of excluded cases involved, it is imperative to compare the differences between the original and reduced sample population.  Furthermore, you need to explain how the sample weight was derived and used in the study.
  4. Psychometric Properties of Scales Used:  Several scales were indicated, but no report on the psychometric properties was documented.
  5. Culture or Ethnic Background:  The cultural diversity, not just the place of origin, may be a contributing factor. Perhaps, this could be elaborated in the section of "Limitations".

Overall, this is an important study that could be further improved in the revision

Author Response

(The authors gave the same response as above.)

Round 2

Reviewer 1 Report

The manuscript entitled “Mental health and traumatization of newly arrived asylum seeker adults in Finland: A population-based study” presents interesting issue, but results which are presented in tables should not be extensively described in the text

Author Response

Thank you again for the comments. We have addressed the suggested changes in the revised manuscript and provide a summary in the attached document.

Reviewer 2 Report

Thank you for the revisions.

Specific comments:

  1. "Self-reported previous mental health diagnosis ..." - these measures may lack credibility due to biased responding. There is the possibility of discordance between self-reported and clinically diagnosed psychiatric disorders as well.
  2. Some concrete suggestions for future research should be stated in the conclusion as well.

Author Response

Thank you again for the comments. We have addressed each of the points in the revised manuscript and provide a summary in the attached document.
